# RISP: Rendering-Invariant State Predictor with Differentiable Simulation and Rendering for Cross-Domain Parameter Estimation

**Pingchuan Ma**[*]
MIT CSAIL
pcma@csail.mit.edu

**Tao Du**[*]
MIT CSAIL
taodu@csail.mit.edu

**Joshua B. Tenenbaum**
MIT BCS, CBMM, CSAIL
jbt@mit.edu

**Wojciech Matusik**
MIT CSAIL
wojciech@csail.mit.edu

**Chuang Gan**
MIT-IBM Watson AI Lab
ganchuang@csail.mit.edu

## Abstract

This work considers identifying parameters characterizing a physical system's dynamic motion directly from a video whose rendering configurations are inaccessible. Existing solutions require massive training data or lack generalizability to unknown rendering configurations. We propose a novel approach that marries domain randomization and differentiable rendering gradients to address this problem. Our core idea is to train a rendering-invariant state-prediction (RISP) network that transforms image differences into state differences independent of rendering configurations, e.g., lighting, shadows, or material reflectance. To train this predictor, we formulate a new loss on rendering variances using gradients from differentiable rendering. Moreover, we present an efficient, second-order method to compute the gradients of this loss, allowing it to be integrated seamlessly into modern deep learning frameworks. We evaluate our method in rigid-body and deformable-body simulation environments using four tasks: state estimation, system identification, imitation learning, and visuomotor control. We further demonstrate the efficacy of our approach on a real-world example: inferring the state and action sequences of a quadrotor from a video of its motion sequences. Compared with existing methods, our approach achieves significantly lower reconstruction errors and has better generalizability among unknown rendering configurations[1].

## 1 Introduction

Reconstructing dynamic information about a physical system directly from a video has received considerable attention in the robotics, machine learning, computer vision, and graphics communities. This problem is fundamentally challenging because of its deep coupling among physics, geometry, and perception of a system. Traditional solutions like motion capture systems (Vicon; OptiTrack; Qualisys) can provide high-quality results but require prohibitively expensive external hardware platforms. More recent development in differentiable simulation and rendering provides an inexpensive and attractive alternative to the motion capture systems and has shown promising proof-of-concept results (Murthy et al., 2020). However, existing methods in this direction typically assume the videos come from a *known* renderer. Such an assumption limits their usefulness in inferring dynamic information from an *unknown* rendering domain, which is common in real-world applications due to the discrepancy between rendering and real-world videos. Existing techniques for aligning different rendering domains, e.g., CycleGAN (Zhu et al., 2017), may help alleviate this issue, but they typically require access to the target domain with massive data, which is not always available. To our best knowledge, inferring dynamic parameters of a physical system directly from videos under *unknown* rendering conditions remains far from being solved, and our work aims to fill this gap.

---

[*]Equal contribution
[1]Videos, code, and data are available on the project webpage: http://risp.csail.mit.edu

Quadrotor          Cube          Hand          Rod

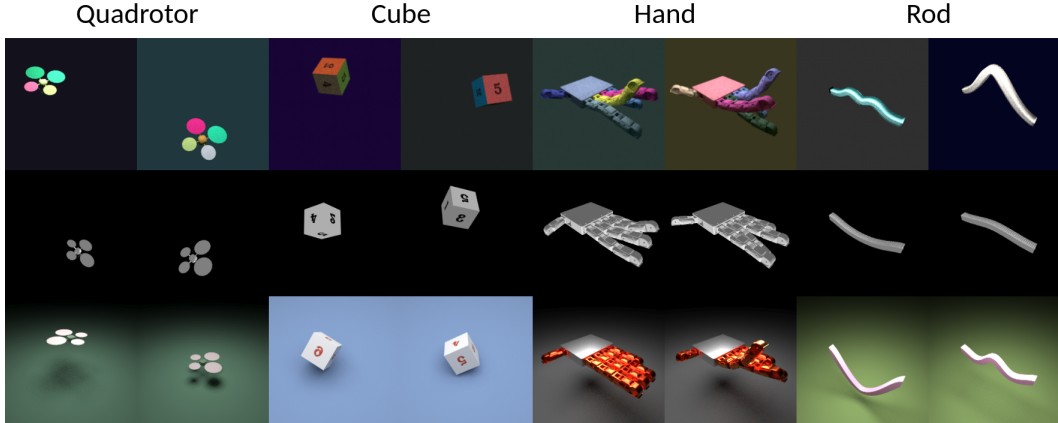

Figure 1: A gallery of our four environments (left to right) across three rendering domains (top to bottom). For each environment, we train a RISP with images under varying lighting, background, and materials generated from a differentiable render (top). Each environment then aims to find proper system and control parameters to simulate and render the physical system (middle) so that it matches the dynamic motion of a reference video (bottom) with unknown rendering configurations. We deliberately let three rows use renderers with vastly different rendering configurations.

We propose a novel approach combining three ideas to address this challenge: domain randomization, state estimation, and rendering gradients. Domain randomization is a classic technique for transferring knowledge between domains by generating massive samples whose variances can cover the discrepancy between domains. We upgrade it with two key innovations: First, we notice that image differences are sensitive to changes in rendering configurations, which shadows the rendering-invariant, dynamics-related parameters that we genuinely aim to infer. This observation motivates us to propose a *rendering-invariant state predictor* (RISP) that extracts state information of a physical system from videos. Our second innovation is to leverage *rendering gradients* from a differentiable renderer. Essentially, requiring the output of RISP to be agnostic to rendering configurations equals enforcing its gradients for rendering parameters to be zero. We propose a new loss function using rendering gradients and show an efficient method for integrating it into deep learning frameworks.

Putting all these ideas together, we develop a powerful pipeline that effectively infers parameters of a physical system directly from video input under random rendering configurations. We demonstrate the efficacy of our approach on a variety of challenging tasks evaluated in four environments (Sec. 4 and Fig. 1) as well as in a real-world application (Fig. 4). The experimental results show that our approach outperforms the state-of-the-art techniques by a large margin in most of these tasks due to the inclusion of rendering gradients in the training process.

In summary, our work makes the following contributions:

- We investigate and identify the bottleneck in inferring state, system, and control parameters of physical systems from videos under various rendering configurations (Sec. 3.1);
- We propose a novel solution combining domain randomization, state estimation, and rendering gradients to achieve generalizability across rendering domains (Sec. 3.2);
- We demonstrate the efficacy of our approach on several challenging tasks in both simulation and real-world environments (Sec. 4).

## 2 RELATED WORK

**Differentiable simulation**   Differentiable simulation equips traditional simulation with gradient information. Such additional gradient information connects simulation tasks with numerical optimization techniques. Previous works have demonstrated the power of gradients from differentiable simulators in rigid-body dynamics (Geilinger et al., 2020; Degrave et al., 2019; de Avila Belbute-Peres et al., 2018; Xu et al., 2021; Hong et al., 2021; Qiao et al., 2021a), deformable-body dynamics (Du et al., 2021b; Huang et al., 2020; Du et al., 2021a; Hu et al., 2019b; Gan et al., 2021; Hahn et al.,

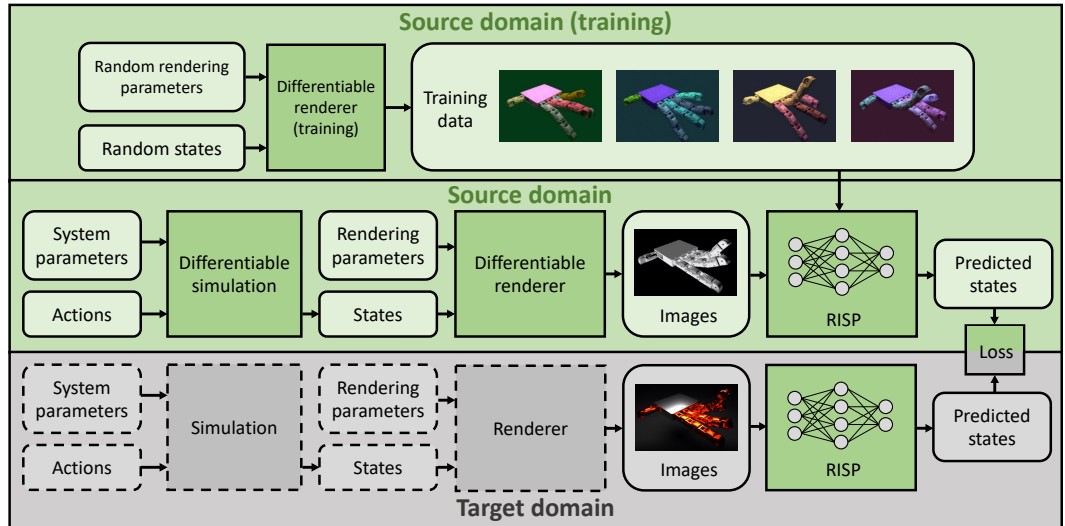

Figure 2: An overview of our method (Sec. 3). We first train RISP using images rendered with random states and rendering parameters (top). We then append RISP to the output of a differentiable renderer, leading to a fully differentiable pipeline from system and control parameters to states predicted from images (middle). Given reference images generated from unknown parameters (dashed gray boxes) in the target domain (bottom), we feed them to RISP and minimize the discrepancies between predicted states (rightmost green-gray box) to reconstruct the underlying system parameters, states, or actions.

2019; Ma et al., 2021; Qiao et al., 2021b), fluids (Du et al., 2020; McNamara et al., 2004; Hu et al., 2019a), and co-dimensional objects (Qiao et al., 2020; Liang et al., 2019). We make heavy use of differentiable simulators in this work but our contribution is orthogonal to them: we treat differentiable simulation as a black box, and our proposed approach is agnostic to the choice of simulators.

**Differentiable rendering**  Differentiable rendering offers gradients for rendering inputs, e.g., lighting, materials, or shapes (Ramamoorthi et al., 2007; Li et al., 2015; Jarosz et al., 2012). The state-of-the-art differentiable renderers (Li et al., 2018; Nimier-David et al., 2019) are powerful in handling gradients even with global illumination or occlusion. Our work leverages these renderers but with a different focus on using their gradients as a physics prior in a learning pipeline.

**Domain randomization**  The intuition behind domain randomization (Tobin et al., 2017; Peng et al., 2018; Andrychowicz et al., 2020; Sadeghi & Levine, 2017; Tan et al., 2018) is that a model can hopefully cross the domain discrepancy by seeing a large amount of random data in the source domain. This often leads to robust but conservative performance in the target domain. The generalizability of domain randomization comes from a more *robust* model that attempts to absorb domain discrepancies by behaving conservatively, while the generalizability of our method comes from a more *accurate* model that aims to match first-order gradient information.

## 3  METHOD

Given a video showing the dynamic motion of a physical system, our goal is to infer the unknown state, system, or control parameters directly from the video with partial knowledge about the physics model and rendering conditions. Specifically, we assume we know the governing equations of the physical system (e.g., Newton's law for rigid-body systems) and the camera position, but the exact system, control, or rendering parameters are not exposed.

To solve this problem, we propose a pipeline (Fig. 2) that consists of two components: 1) a differentiable simulation and rendering engine; 2) the RISP network. First, we use our engine to simulate and render the state of a physical system and outputs images under varying rendering configuration. Next, the RISP network learns to reconstruct the state information from these generated images. Putting

these two components together, we have a pipeline that can faithfully recover dynamic information of a physical system from a new video with unseen rendering configurations.

## 3.1 DIFFERENTIABLE SIMULATION AND RENDERING ENGINE

Given a physical system with known dynamic model $\mathcal{M}$, we first use a differentiable simulator to simulate its states based on action inputs at each time step after time discretization:

$$\mathbf{s}_{i+1} = \mathcal{M}_{\phi}(\mathbf{s}_i, \mathbf{a}_i), \quad \forall i = 0, 1, \cdots, N-1, \tag{1}$$

where $N$ is the number of time steps in a rollout of physics simulation, and $\mathbf{s}_i$, $\mathbf{s}_{i+1}$ and $\mathbf{a}_i$ represent the state and action vectors at the corresponding time steps, respectively. The $\phi$ vector encodes the system parameters in the model, e.g., mass, inertia, and elasticity. Next, we apply a differentiable renderer $\mathcal{R}$ to generate an image $\mathbf{I}_i$ for each state $\mathbf{s}_i$:

$$\mathbf{I}_i = \mathcal{R}_{\psi}(\mathbf{s}_i), \quad \forall i = 0, 1, \cdots, N. \tag{2}$$

Here, $\psi$ is a vector encoding rendering parameters whose gradients are available in the renderer $\mathcal{R}$. Examples of $\psi$ include light intensity, material reflectance, or background color. By abuse of notation, we re-write the workflow of our simulation and rendering engine to a compact form:

$$\{\mathbf{I}_i\} = \mathcal{R}_{\psi}[\underbrace{\mathcal{M}_{\phi}(\mathbf{s}_0, \{\mathbf{a}_i\})}_{\{\mathbf{s}_i\}}]. \tag{3}$$

In other words, given an initial state $\mathbf{s}_0$ and a sequence of actions $\{\mathbf{a}_i\}$, we generate a sequence of states $\{\mathbf{s}_i\}$ from simulation and renders the corresponding image sequence $\{\mathbf{I}_i\}$. The task of recovering unknown information from a reference video $\{\mathbf{I}_i^{\text{ref}}\}$ can be formulated as follows:

$$\min_{\mathbf{s}_0, \{\mathbf{a}_i\}, \phi, \psi} \quad \mathcal{L}(\{\mathbf{I}_i^{\text{ref}}\}, \{\mathbf{I}_i\}), \tag{4}$$

$$\text{s.t.} \quad \{\mathbf{I}_i\} = \mathcal{R}_{\psi}[\mathcal{M}_{\phi}(\mathbf{s}_0, \{\mathbf{a}_i\})], \tag{5}$$

where $\mathcal{L}$ is a loss function penalizing the difference between the generated images and their references. Assuming that the simulator $\mathcal{M}$ and the renderer $\mathcal{R}$ are differentiable with respect to their inputs, we can run gradient-based optimization algorithms to solve Eqn. (4). This is essentially the idea proposed in $\nabla$Sim, the state-of-the-art method for identifying parameters directly from video inputs (Murthy et al., 2020). Specifically, $\nabla$Sim defines $\mathcal{L}$ as a norm on pixelwise differences.

One major limitation in Eqn. (4) is that it expects reasonably similar initial images $\{\mathbf{I}_i\}$ and references $\{\mathbf{I}_i^{\text{ref}}\}$ to successfully solve the optimization problem. Indeed, since the optimization problem is highly nonlinear due to its coupling between simulation and rendering, local optimization techniques like gradient-descent can be trapped into local minima easily if $\{\mathbf{I}_i\}$ and $\{\mathbf{I}_i^{\text{ref}}\}$ are not close enough. While $\nabla$Sim has reported promising results when $\{\mathbf{I}_i\}$ and $\{\mathbf{I}_i^{\text{ref}}\}$ are rendered with moderately different $\psi$, we found in our experiments that directly optimizing $\mathcal{L}$ defined on the image space rarely works when the two rendering domains are vastly different (Fig. 1). Therefore, we believe it requires a fundamentally different solution, motivating us to propose RISP in our method.

## 3.2 THE RISP NETWORK

The difficulty of generalizing Eqn. (4) across different rendering domains is partially explained by the fact that the loss $\mathcal{L}$ is defined on the differences in the image space, which is sensitive to changes in rendering configurations. To address this issue, we notice from many differentiable simulation papers that a loss function in the state space is fairly robust to random initialization (Du et al., 2020; Liang et al., 2019), inspiring us to redefine $\mathcal{L}$ in a state-like space. More concretely, we introduce the RISP network $\mathcal{N}$ that takes as input an image $\mathbf{I}$ and outputs a state prediction $\hat{\mathbf{s}} = \mathcal{N}(\mathbf{I})$. We then redefine the optimization problem in Eqn. (4) as follows (Fig. 2):

$$\min_{\mathbf{s}_0, \{\mathbf{a}_i\}, \phi, \psi} \quad \mathcal{L}(\mathcal{N}_{\boldsymbol{\theta}}(\{\mathbf{I}_i^{\text{ref}}\}), \mathcal{N}_{\boldsymbol{\theta}}(\{\mathbf{I}_i\})), \tag{6}$$

$$\text{s.t.} \quad \{\mathbf{I}_i\} = \mathcal{R}_{\psi}[\mathcal{M}_{\phi}(\mathbf{s}_0, \{\mathbf{a}_i\})]. \tag{7}$$

Note that the network $\mathcal{N}_{\boldsymbol{\theta}}$, parametrized by $\boldsymbol{\theta}$, is pre-trained and fixed in this optimization problem. Essentially, Eqn. (6) maps the two image sequences to the predicted state space, after which the

standard gradient-descent optimization follows. A well-trained network $\mathcal{N}$ can be interpreted as an "inverse renderer" $\mathcal{R}^{-1}$ that recovers the rendering-invariant state vector regardless of the choice of rendering parameters $\boldsymbol{\psi}$, allowing Eqn. (6) to match the information behind two image sequences $\{\mathbf{I}_i\}$ and $\{\mathbf{I}_i^{\text{ref}}\}$ even when they are generated from different renderers $\mathcal{R}_{\boldsymbol{\psi}}$. Below, we present two ideas to train the network $\mathcal{N}$:

**The first idea: domain randomization**  Our first idea is to massively sample state-rendering pairs $(\mathbf{s}_j, \boldsymbol{\psi}_j)$ and render the corresponding image $\mathbf{I}_j = \mathcal{R}_{\boldsymbol{\psi}_j}(\mathbf{s}_j)$, giving us a training set $\mathcal{D} = \{(\mathbf{s}_j, \boldsymbol{\psi}_j, \mathbf{I}_j)\}$. We then train $\mathcal{N}$ to minimize the prediction error:

$$\mathcal{L}^{\text{error}}(\boldsymbol{\theta}, \mathcal{D}) = \sum_{(\mathbf{s}_j, \boldsymbol{\psi}_j, \mathbf{I}_j) \in \mathcal{D}} \underbrace{\|\mathbf{s}_j - \mathcal{N}_{\boldsymbol{\theta}}(\mathbf{I}_j)\|_1}_{\mathcal{L}_j^{\text{error}}}. \tag{8}$$

The intuition is straightforward: $\mathcal{N}_{\boldsymbol{\theta}}$ learns to generalize over rendering configurations because it sees images generated with various rendering parameters $\boldsymbol{\psi}$. This is exactly the domain randomization idea (Tobin et al., 2017), which we borrow to solve our problem across different rendering domains.

**The second idea: rendering gradients**  One major bottleneck in domain randomization is its needs for massive training data that spans the whole distribution of rendering parameters $\boldsymbol{\psi}$. Noting that a perfectly rendering-invariant $\mathcal{N}$ must satisfy the following condition:

$$\frac{\partial \mathcal{N}_{\boldsymbol{\theta}}(\mathcal{R}_{\boldsymbol{\psi}}(\mathbf{s}))}{\partial \boldsymbol{\psi}} \equiv \mathbf{0}, \quad \forall \mathbf{s}, \boldsymbol{\psi}, \tag{9}$$

we consider adding a regularizer to the training loss:

$$\mathcal{L}^{\text{train}}(\boldsymbol{\theta}, \mathcal{D}) = \mathcal{L}^{\text{error}} + \gamma \underbrace{\sum_{(\mathbf{s}_j, \boldsymbol{\psi}_j, \mathbf{I}_j) \in \mathcal{D}} \|\frac{\partial \mathcal{N}_{\boldsymbol{\theta}}(\mathcal{R}_{\boldsymbol{\psi}_j}(\mathbf{s}_j))}{\partial \boldsymbol{\psi}_j}\|_{\text{F}}}_{\mathcal{L}^{\text{reg}}}, \tag{10}$$

where $\|\cdot\|_{\text{F}}$ indicates the Frobenius norm and $\gamma$ a regularization weight. The intuition is that by suppressing this Jacobian to zero, we encourage the network $\mathcal{N}$ to flatten out its landscape along the dimension of rendering parameters $\boldsymbol{\psi}$, and invariance across rendering configurations follows. To implement this loss, we apply the chain rule:

$$\frac{\partial \mathcal{N}_{\boldsymbol{\theta}}(\mathcal{R}_{\boldsymbol{\psi}_j}(\mathbf{s}_j))}{\partial \boldsymbol{\psi}_j} = \frac{\partial \mathcal{N}_{\boldsymbol{\theta}}(\mathbf{I}_j)}{\partial \boldsymbol{\psi}_j} = \frac{\partial \mathcal{N}_{\boldsymbol{\theta}}(\mathbf{I}_j)}{\partial \mathbf{I}_j} \frac{\partial \mathbf{I}_j}{\partial \boldsymbol{\psi}_j}, \tag{11}$$

where the first term $\frac{\partial \mathcal{N}_{\boldsymbol{\theta}}(\mathbf{I}_j)}{\partial \mathbf{I}_j}$ is available in any modern deep learning frameworks and the second term $\frac{\partial \mathbf{I}_j}{\partial \boldsymbol{\psi}_j}$ can be obtained from the state-of-the-art differentiable renderer (Nimier-David et al., 2019). We can now see more clearly the intuition behind RISP: it requires the network's sensitivity about input images to be orthogonal to the direction that rendering parameters can influence the image, leading to a rendering-invariant prediction.

We stress that the design of this new loss in Eqn. (10) is non-trivial. In fact, both $\mathcal{L}^{\text{error}}$ and $\mathcal{L}^{\text{reg}}$ have their unique purposes and must be combined: $\mathcal{L}^{\text{error}}$ encourages $\mathcal{N}$ to fit its output to individually different states, and $\mathcal{L}^{\text{reg}}$ attempts to smooth out its output along the $\boldsymbol{\psi}$ dimension. Specifically, $\mathcal{L}^{\text{reg}}$ cannot be optimized as a standalone loss because it leads to a trivial solution of $\mathcal{N}$ always predicting constant states. Putting $\mathcal{L}^{\text{error}}$ and $\mathcal{L}^{\text{reg}}$ together forces them to strike a balance between predicting accurate states and ignoring noises from rendering conditions, leading to a network $\mathcal{N}$ that truly learns the "inverse renderer" $\mathcal{R}^{-1}$.

It remains to show how to compute the gradient of the regularizer $\mathcal{L}^{\text{reg}}$ with respect to the network parameters $\boldsymbol{\theta}$, which is required by gradient-based optimizers to minimize this new loss. As the loss definition now includes first-order derivatives, computing its gradients involves second-order partial derivatives, which can be time-consuming if implemented carelessly with multiple loops. Our last contribution is to provide an efficient method for computing this gradient, which can be fully implemented with existing frameworks (PyTorch and `mitsuba-2` in our experiments):

**Theorem 1** *Assuming forward mode differentiation is available in the renderer $\mathcal{R}$ and reverse mode differentiation is available in the network $\mathcal{N}$, we can compute a stochastic gradient $\frac{\partial \mathcal{L}^{\text{reg}}}{\partial \boldsymbol{\theta}}$ in $\mathcal{O}(|\mathbf{s}||\boldsymbol{\theta}|)$ time per image using pre-computed data occupying $\mathcal{O}(\sum_j |\boldsymbol{\psi}_j||\mathbf{I}_j|)$ space.*

In particular, we stress that computing the gradients of $\mathcal{L}^{\text{reg}}$ does *not* require second-order gradients in the renderer $\mathcal{R}$, which would exceed the capability of all existing differentiable renderers we are aware of. We leave the proof of this theorem in our supplemental material.

**Further speedup** Theorem 1 states that it takes time linear to the network size and state dimension to compute the gradients of $\mathcal{L}^{\text{reg}}$. The $\mathcal{O}(|\mathbf{s}||\boldsymbol{\theta}|)$ time cost is affordable for very small rigid-body systems (e.g., $|\mathbf{s}| < 10$) but not scalable for larger systems. Therefore, we use a slightly different regularizer in our implementation:

$$\mathcal{L}^{\text{train}}(\boldsymbol{\theta}, \mathcal{D}) = \mathcal{L}^{\text{error}} + \gamma \sum_{(\mathbf{s}_j, \boldsymbol{\psi}_j, \mathbf{I}_j) \in \mathcal{D}} \|\frac{\partial \mathcal{L}_j^{\text{error}}}{\partial \boldsymbol{\psi}_j}\|. \tag{12}$$

In other words, we instead encourage the state prediction error to be rendering-invariant. It can be seen from the proof in Theorem 1 that this new regularizer requires only $\mathcal{O}(\boldsymbol{\theta})$ time to compute its gradients, and we have found empirically that the performance of this new regularizer is comparable to Eqn. (10) but is much faster. We leave a theoretical analysis of the two regularizers to future work.

## 4 EXPERIMENTS

In this section, we conduct various experiments to study the following questions:

- Q1: Is pixelwise loss on videos across rendering domains sufficient for parameter prediction?
- Q2: If pixelwise loss is not good enough, are there other competitive alternatives to the state-prediction loss in our approach?
- Q3: Is the regularizer on rendering gradients in our loss necessary?
- Q4: How does our approach compare with directly optimizing state discrepancies?
- Q5: Is the proposed method applicable to real-world scenarios?

We address the first four questions using the simulation environment described in Sec. 4.1 and answer the last question using a real-world application at the end of this section. More details about the experiments, including ablation study, can be found in Appendix.

### 4.1 EXPERIMENTAL SETUP

**Environments** We implement four environments (Fig. 1): a rigid-body environment without contact (**quadrotor**), a rigid-body environment with contact (**cube**), an articulated body (**hand**), and a deformable-body environment (**rod**). Each environment contains a differentiable simulator (Xu et al., 2021; Du et al., 2021b) and a differentiable renderer (Li et al., 2018). We deliberately generated the training set in Sec. 3 using a different renderer (Nimier-David et al., 2019) and used different distributions when sampling rendering configurations in the training set and the environments.

**Tasks** We consider four types of tasks defined on the physical systems in all environments: state estimation, system identification, imitation learning, and visuomotor control. The state estimation task (Sec. 4.2) require a model to predict the state of the physical system from a given image and serves as a prerequisite for the other downstream tasks. The system identification (Sec. 4.3) and imitation learning (Sec. 4.4) tasks aim to recover the system parameters and control signals of a physical system from the video, respectively. Finally, in the visuomotor control task (Appendix), we replace the video with a target image showing the desired state of the physical system and aim to discover the proper control signals that steer the system to the desired state. In all tasks, we use a photorealistic renderer (Pharr et al., 2016) to generate the target video or image.

**Baselines** We consider two strong baselines: The **pixelwise-loss** baseline is used by $\nabla\text{Sim}$ (Murthy et al., 2020), which is the state-of-the-art method for identifying system parameters directly from video inputs. We implement $\nabla\text{Sim}$ by removing RISP from our method and backpropagating the pixelwise loss on images through differentiable rendering and simulation. We run this baseline to analyze the limit of pixelwise loss in downstream tasks (Q1). The second baseline is **preceptual-loss** (Johnson et al., 2016), which replaces the pixelwise loss in $\nabla\text{Sim}$ with loss functions based on high-level features extracted by a pre-trained CNN. By comparing this baseline with our method, we can justify why we choose to let RISP predict states instead of other percepual features (Q2).

We also include two weak baselines used by $\nabla \texttt{Sim}$: The **average** baseline is a deterministic method that always returns the average quantity observed from the training set, and the **random** baseline returns a guess randomly drawn from the data distribution used to generate the training set. We use these two weak baselines to avoid designing environments and tasks that are too trivial to solve.

**Our methods** We consider two versions of our methods in Sec. 3: **ours-no-grad** implements the domain randomization idea without using the proposed regularizers, and **ours** is the full approach that includes the regularizer using rendering gradients. By comparing between them, we aim to better understand the value of the rendering gradients in our proposed method (Q3).

**Oracle** Throughout our experiments, we also consider an oracle method that directly minimizes the state differences obtained from simulation outputs without further rendering. In particular, this oracle sees the ground-truth states for each image in the target video or image, which is inaccessible to all baselines and our methods. We consider this approach to be an oracle because it is a subset of our approach that involve differentiable physics only, but it needs a perfect state-prediction network. This oracle can give us an upper bound for the performance of our method (Q4).

**Training** We build our RISP network and baselines upon a modified version of ResNet-18 (He et al., 2016) and train them with the Adam optimizer (Kingma & Ba, 2014). We report more details of our network architecture, training strategies, and hyperparameters in Appendix.

## 4.2 State Estimation Results

|              | quadrotor | cube | hand | rod |
|:---:|:---:|:---:|:---:|:---:|
| average      | $0.5994 \pm 0.0000$ | $0.5920 \pm 0.0000$ | $0.2605 \pm 0.0000$ | $0.9792 \pm 0.0000$ |
| random       | $0.9661 \pm 0.7548$ | $0.9655 \pm 0.8119$ | $0.8323 \pm 0.5705$ | $1.2730 \pm 0.7197$ |
| ours-no-grad | $0.3114 \pm 0.3191$ | $0.2805 \pm 0.3199$ | $0.1155 \pm 0.0505$ | $0.0201 \pm 0.0087$ |
| ours         | $\mathbf{0.1505 \pm 0.1163}$ | $\mathbf{0.1642 \pm 0.1887}$ | $\mathbf{0.0974 \pm 0.0255}$ | $\mathbf{0.0194 \pm 0.0048}$ |

Table 1: State estimation results (Sec. 4.2). Each entry in the table reports the mean and standard deviation of the state estimation error computed from 800 images under 4 rendering configurations.

In this task, we predict the states of the physical system from randomly generated images and report the mean and standard deviation of the state prediction errors in Table 1. Note that we exclude the **perceptual-loss** and **pixelwise-loss** baselines as they do not require a state prediction step. Overall, we find that the state estimation results from our methods are significantly better than all baselines across the board. The two weak baselines perform poorly, confirming that this state-estimation task cannot be solved trivially. We highlight that our method with the rendering-gradient loss predicts the most stable and accurate state of the physical system across the board, which strongly demonstrates that RISP learns to make predictions independent of various rendering configurations.

## 4.3 System Identification Results

Our system identification task aims to predict the system parameters of a physical system, e.g., mass, density, stiffness, or elasticity, by watching a reference video with known action sequences. For each environment, we manually design a sequence of actions and render a reference video of its dynamic motion. Next, we randomly pick an initial guess of the system parameters and run gradient-based optimization using all baselines, our methods, and the oracle. We repeat this experiment 4 times with randomly generated rendering conditions and initial guesses and report the mean and standard deviation of each system parameter in Table 2. The near-perfect performance of the oracle suggests that these system identification tasks are feasible to solve as long as a reliable state estimation is available. Both of our methods outperform almost all baselines by a large margin, sometimes even by orders of magnitude. This is as expected, since the previous task already suggests that our methods can predict states from a target video much more accurate than baselines, which is a crucial prerequisite for solving system identification. The only exception is in the **cube** environment, where the pixelwise loss performs surprisingly well. We hypothesize it may be due to its relatively simple geometry and high contrast from the background (Fig. 1).

|  | **quadrotor** mass | **cube** stiffness | **hand** joint stiffness | **rod** Young's modulus |
|---|---|---|---|---|
| random | 9.22e-2±3.83e-2 | 2.31e-1±9.57e-2 | 5.70e-1±1.62e-1 | 1.30e6±1.35e6 |
| pixelwise-loss | 7.22e-2±5.26e-2 | **2.24e-3±1.74e-3** | 1.04e-1±9.62e-2 | 8.50e5±1.42e6 |
| perceptual-loss | 6.45e-2±5.23e-2 | 1.16e-1±5.83e-2 | 1.10e-1±1.18e-1 | 8.32e5±1.43e6 |
| ours-no-grad | 6.07e-2±4.34e-2 | 1.16e-1±6.20e-2 | 4.85e-2±**2.16e-2** | 8.78e4±1.52e5 |
| ours | **1.18e-2±1.93e-2** | 6.76e-3±7.23e-3 | **3.96e-2**±2.73e-2 | **9.31e1±4.21e1** |
| oracle | 2.36e-5±2.41e-5 | 1.15e-3±8.60e-4 | 3.92e-3±4.40e-4 | 4.36e0±3.57e0 |

Table 2: System identification results (Sec. 4.3). Each entry reports the mean and standard deviation of the parameter estimation error computed from 4 random initial guesses and rendering conditions.

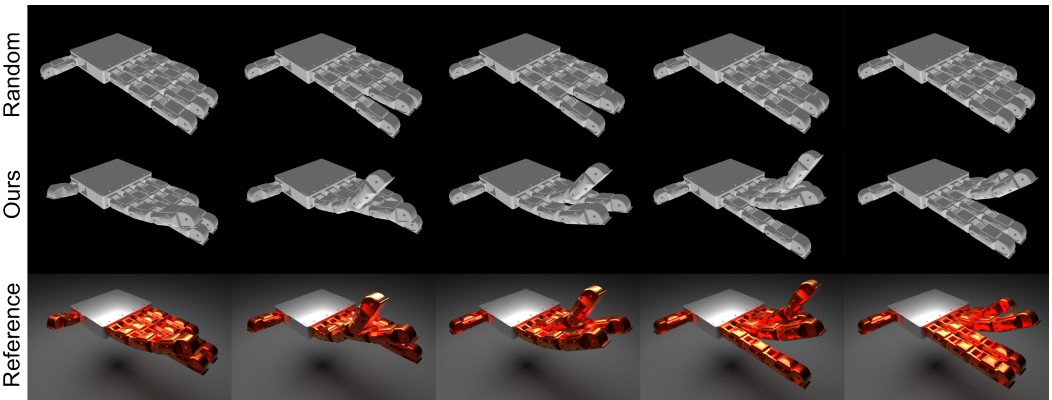

Figure 3: Imitation learning in the **hand** environment (Sec. 4.4). Given a reference video (bottom row, shown as five intermediate frames), the goal is to reconstruct a sequence of actions that resembles its motion. We show the motions generated using a randomly chosen initial guess of the actions (top row) and optimized actions using our method with rendering gradients (middle row).

## 4.4 IMITATION LEARNING RESULTS

Our imitation learning tasks consider the problem of emulating the dynamic motion of a reference video. The experiment setup is similar to the system identification task except that we swap the known and unknown variables in the environment: The system parameters are now known, and the goal is to infer the unknown sequence of actions from the reference video. Note that the **cube** environment is excluded because it has no control signals. As before, we repeat the experiment in all environments 4 times with randomly generated rendering configurations and initial guesses of the actions. We report the results in Table 3 and find that our method with rendering gradients (**ours** in the table) achieves much lower errors, indicating that we resemble the motions in the video much more accurately. The errors from **pixelwise-loss** have smaller variations across rendering domains but are larger than ours, indicating that it struggles to solve this task under all four rendering configurations. In addition, the oracle finds a sequence of actions leading to more similar motions than our method, but it requires full and accurate knowledge of the state information which is rarely accessible from a video. We visualize our results in the **hand** environment in Fig. 3.

## 4.5 A REAL-WORLD APPLICATION

Finally, we evaluate the efficacy of our approach on a real-world application: Given a video of a flying quadrotor, we aim to build its digital twin that replicates the video motion by inferring a reasonable sequence of actions. This real-to-sim transfer problem is challenging due to a few reasons: First, the real-world video contains complex textures, materials, and lighting conditions unknown to us and unseen by our differentiable renderer. Second, the quadrotor's real-world dynamics is polluted by environmental noises and intricate aerodynamic effects, which are ignored in our differentiable simulation environment. Despite these challenges, our method achieved a qualitatively good result with a standard differentiable rigid-body simulator and renderer, showing its generalizability across

|  | quadrotor | hand | rod |
|---|---|---|---|
| average | $28.40 \pm 0.00$ | $7.62 \pm 0.00$ | $30.20 \pm 0.00$ |
| random | $1120 \pm 113$ | $10.27 \pm 0.91$ | $29.89 \pm 0.61$ |
| pixelwise-loss | $12.65 \pm \mathbf{0.13}$ | $6.71 \pm \mathbf{0.07}$ | $29.57 \pm 0.85$ |
| perceptual-loss | N/A | $5.10 \pm 1.80$ | $14.61 \pm 5.13$ |
| ours-no-grad | $25.07 \pm 5.76$ | $7.12 \pm 0.71$ | $\mathbf{0.83} \pm 0.35$ |
| ours | $\mathbf{2.63} \pm 1.86$ | $\mathbf{1.52} \pm 0.18$ | $1.05 \pm \mathbf{0.27}$ |
| oracle | $0.79 \pm 0.44$ | $0.02 \pm 0.02$ | $0.28 \pm 0.16$ |

Table 3: Imitation learning results (Sec. 4.4). Each entry reports the mean and standard deviation of the state discrepancy computed from 4 randomly generated initial guesses and rendering conditions. N/A indicates failure of convergence after optimization.

moderate discrepancies in dynamic models and rendering configurations (Fig. 4). Our approach outperforms all baselines by a large margin in this task, which we detail in Appendix.

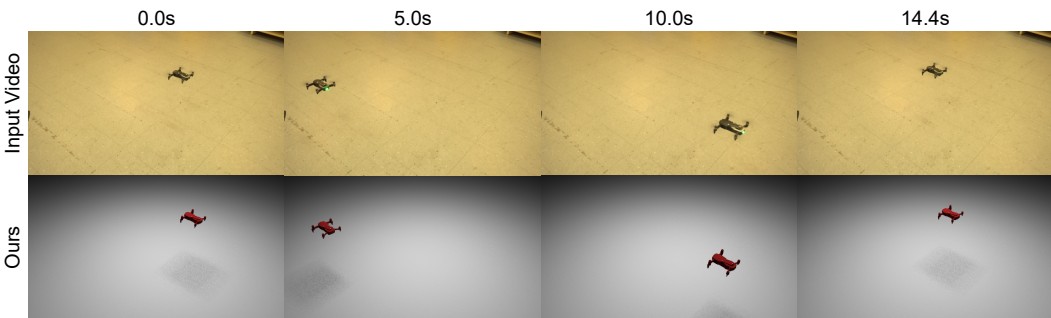

Figure 4: Imitation learning in the real-world experiment (Sec. 4.5). Given a reference video (top row), the goal is to reconstruct a sequence of actions sent to a virtual quadrotor that resembles its motion. We illustrate the motions reconstructed using our method (bottom row).

## 5 CONCLUSIONS, LIMITATIONS, AND FUTURE WORK

We proposed a framework that integrates rendering-invariant state-prediction into differentiable simulation and rendering for cross-domain parameter estimation. The experiments in simulation and in real world have shown that our method is more robust than pixel-wise or perceptual losses on unseen rendering configurations. The additional ablated study further confirms the efficiency and efficacy comes from using the rendering gradients in our RISP network.

Despite its promising results, RISP still has a few limitations. Firstly, we require a differentiable simulator that can capture the dynamic model of the object, which may not be available for real-world scenes with intricate dynamics, e.g., fluids. A second limitation is that our method requires knowledge of camera parameters, which is not always accessible in a real-world scenario. This limitation can be resolved by incorporating in RISP the gradients for camera parameters available in modern differentiable renderers. Lastly, we assume a moderately accurate object geometry is available, which can potentially be relaxed by combining NeRF (Mildenhall et al., 2020) with our approach to infer the geometry.

## ACKNOWLEDGMENTS

We thank Sai Praveen Bangaru for our discussions on differentiable rendering. This work was supported by MIT-IBM Watson AI Lab and its member company Nexplore, ONR MURI, DARPA Machine Common Sense program, ONR (N00014-18-1-2847), and Mitsubishi Electric.

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

# A  PROOF OF THE THEOREM

For brevity, we remove the summation in $\mathcal{L}^{\mathrm{reg}}$ and drop the index $j$ to focus on deriving the gradients of the Frobenius term with respect to the network parameters $\boldsymbol{\theta}$. Let $\mathbf{G}$ be the Jacobian matrix inside the Frobenius norm, and let $i$ and $j$ be its row and column indices. We now derive the gradient of $\mathcal{L}^{\mathrm{reg}}$ with respect to the $k$-th parameter in $\boldsymbol{\theta}$ as follows:

$$\frac{\partial \mathcal{L}^{\mathrm{reg}}}{\partial \boldsymbol{\theta}_k} = \sum_{ij} 2\mathbf{G}_{ij} \frac{\partial \mathbf{G}_{ij}}{\partial \boldsymbol{\theta}_k} \tag{13}$$

$$= \sum_{ij} 2\mathbf{G}_{ij} \frac{\partial}{\partial \boldsymbol{\theta}_k} \left[ \frac{\partial \mathcal{N}_{\boldsymbol{\theta}}(\mathbf{I})_i}{\partial \mathbf{I}} : \frac{\partial \mathbf{I}}{\partial \boldsymbol{\psi}_j} \right], \tag{14}$$

$$= \sum_{ij} 2\mathbf{G}_{ij} \left[ \frac{\partial^2 \mathcal{N}_{\boldsymbol{\theta}}(\mathbf{I})_i}{\partial \mathbf{I} \partial \boldsymbol{\theta}_k} : \frac{\partial \mathbf{I}}{\partial \boldsymbol{\psi}_j} \right] \tag{15}$$

$$= \sum_{i} \frac{\partial^2 \mathcal{N}_{\boldsymbol{\theta}}(\mathbf{I})_i}{\partial \mathbf{I} \partial \boldsymbol{\theta}_k} : \left( \sum_{j} 2\mathbf{G}_{ij} \frac{\partial \mathbf{I}}{\partial \boldsymbol{\psi}_j} \right) \tag{16}$$

$$= \sum_{i} \frac{\partial^2 \mathcal{N}_{\boldsymbol{\theta}}(\mathbf{I})_i}{\partial \mathbf{I} \partial \boldsymbol{\theta}_k} : 2 \sum_{j} \left( \frac{\partial \mathcal{N}_{\boldsymbol{\theta}}(\mathbf{I})_i}{\partial \mathbf{I}} : \frac{\partial \mathbf{I}}{\partial \boldsymbol{\psi}_j} \right) \frac{\partial \mathbf{I}}{\partial \boldsymbol{\psi}_j}. \tag{17}$$

The derivation suggests that we can loop over the state dimension (indexed by $i$) and run backpropagation to obtain $\frac{\partial \mathcal{L}^{\mathrm{reg}}}{\partial \boldsymbol{\theta}}$, assuming that all $\frac{\partial \mathbf{I}}{\partial \boldsymbol{\psi}_j}$ are available. In fact, $\frac{\partial \mathbf{I}}{\partial \boldsymbol{\psi}_j}$ depends on the training data only and remains constant throughout the whole optimization process, so we can pre-compute them in the training set, occupying an extra space of $\mathcal{O}(\sum |\boldsymbol{\psi}||\mathbf{I}|)$. Typically, $|\boldsymbol{\psi}_j|$ is a small number (less than 10 in most of our experiments), so it is affordable to store it with the images generated in the data set. This is also why we prefer a differentiable renderer that offers forward mode differentiation.

Assuming all $\frac{\partial \mathbf{I}}{\partial \boldsymbol{\psi}_j}$ are ready to use, we can implement Eqn. (17) by two backpropagations for each state dimension $i$. First, backpropagating through $\mathcal{N}$ to obtain $\frac{\partial \mathcal{N}_{\boldsymbol{\theta}}(\mathbf{I})_i}{\partial \mathbf{I}}$. Next, we use $\frac{\partial \mathbf{I}}{\partial \boldsymbol{\psi}_j}$ to assemble the adjoint vector (the second summation in Eqn. (17)). Finally, we use the adjoint vector to backpropagate again through the second-order partial derivatives $\frac{\partial^2 \mathcal{N}_{\boldsymbol{\theta}}(\mathbf{I})_i}{\partial \mathbf{I} \partial \boldsymbol{\theta}}$. Note that there is no need for looping over $k$ in the second backpropagation. The total time cost is therefore $\mathcal{O}(|\mathbf{s}||\boldsymbol{\theta}|)$.

# B  IMPLEMENTATION DETAILS

## B.1  ENVIRONMENTS

### B.1.1  QUADCOPTOR

**State space**  We define the physical system of **quadcoptor** as a rigid body system without contact. We explicitly mark a task as "failed" when the center of the quadcoptor has an Euclidean distance to the origin $(0, 0, 0)$ of over 1000 at anytime. We define the state of the quadcoptor as its world position $(x, y, z)$ and rotation angles $(\mathrm{yaw}, \mathrm{pitch}, \mathrm{roll})$. To alleviate the negative impact from the discontinuity of rotation angles, we design the output of the network as:

$$(x, y, z, \sin(\mathrm{yaw}), \sin(\mathrm{pitch}), \sin(\mathrm{roll}), \cos(\mathrm{yaw}), \cos(\mathrm{pitch}), \cos(\mathrm{roll})), \tag{18}$$

and later restore the rotation angles by $\mathrm{atan2}$.

**Action space**  The control signals are 4-d representing the torques applied on 4 propellers. The magnitude of control signals is clamped to $[-500, 500]$.

**Parameter space**  In system identification experiment, we uniformly sample the mass of the quadcoptor within $[2.8, 3.2]$. The ground-truth value of the mass is 3.

### B.1.2 CUBE

**State space**  We define the physical system of **cube** as a rigid body system with contact. Similar to **quadcoptor** environment, we design the output of the network as a 9-d vector following 18.

**Parameter space**  In **cube** environment, we adopt a soft contact model paramterized by the spring stiffness $k_n$. In system identification experiment, we train $\lambda_k = \log_{10} k_n$ for better numerical stability and performance. We uniformly sample $\lambda_k$ within $[5.5, 6.5)$. The ground-truth value of $\lambda_k$ is 6.

### B.1.3 HAND

**State space**  We define the physical system of **hand** as an articulated body system powered by (42). We fix the palm on the world frame and model the articulations as revolve joints. There are 13 joints in this environment, and each joints is clamped within $[-\frac{\pi}{4}, \frac{\pi}{4}]$. We define the state space of **hand** as a 13-d vector representing the angles of each joint $\{\theta_{1\dots13}\}$.

**Action space**  The hand is controlled by applying torque on each joint. We paramterize the control signal of each finger at time $t$ by:

$$u_t = m_j \sin\left(2\pi \min\left(1.0, \max\left(0.0, \frac{1}{T}(t - \delta_j)\right)\right)\right),$$

where $0 \leq m_j \leq 1$ and $0 \leq \delta_j \leq 40$ denote the action magnitude and action bias of finger $j$, and $T = 40$ is the period.

**Parameter space**  For each finger, we let the joint stiffness of all joints on the finger available for training, making the parameter space a 5-d space. We uniformly sample the joint stiffness within $[0.0, 1.0)$. The ground-truth values of the joint stiffness are all 1.0.

### B.1.4 ROD

**State space**  We define the physical system of **rod** as a soft-body dynamics system powered by (6). We fix both ends of the rod and apply a gravity force on it. We define a 10-d state space by evenly selecting 10 sensors along the central spine and track their positions along the vertical direction.

**Action space**  We apply an external force on the center of the rob vertically as the action signal. We clamp the action within $[-500, 500)$.

**Parameter space.**  We parameter the **rod** environment by Young's modulus $E$. We train $\lambda_E = \log_{10} E$ for better numerical stability and performance. We uniformly sample $\lambda_E$ within $[4, 7)$ where the ground-truth value of $\lambda_E$ is 5.

### B.2 THE RISP NETWORK

**Network setting**  We build the network upon a modified version of ResNet-18 (10). All layers are followed by instance normalization (40) without affine parameters, following James et al. (15). Since ReLU degenerates second derivatives to constant values, we replace it with *Swish* non-linearities (35). We change the last layer of ResNet-18 backbone to a linear projection layer with the same number of output with the state dimension.

**Training**  We use *Adam* optimizer (18) with a learning rate of 1e-2 and a weight decay of 1e-6 for training. The learning rate has a cosine decay scheduler (22). We train the network for 100 epochs. We set the batch size to 16 by default. To avoid model collapse in the early stage of training, we linearly increase the magnitude coefficient of the rendering gradients from 0 to a environment-specific value: we set it to 1 for **quadrotor**, 20 for **cube**, 100 for **hand**, and 30 for **rod**.

|  | quadrotor | hand | rod |
|---|---|---|---|
| average | $1.38 \pm 0.00$ | $0.23 \pm 0.00$ | $0.89 \pm 0.00$ |
| random | $63.93 \pm 9.44$ | $0.32 \pm 0.06$ | $0.87 \pm 0.06$ |
| pixelwise-loss | $3.02 \pm 1.35$ | $0.25 \pm 0.05$ | $0.05 \pm 0.01$ |
| perceptual-loss | $1.59 \pm 0.26$ | $0.28 \pm 0.05$ | $0.11 \pm 0.06$ |
| ours-no-grad | $1.30 \pm \mathbf{0.15}$ | $0.23 \pm 0.06$ | $\mathbf{0.04} \pm \mathbf{0.0002}$ |
| ours | $\mathbf{0.54} \pm 0.25$ | $\mathbf{0.11} \pm \mathbf{0.02}$ | $\mathbf{0.04} \pm 0.0006$ |
| oracle | $0.15 \pm 0.09$ | $0.002 \pm 0.00$ | $0.01 \pm 0.02$ |

Table 4: Visuomotor control results. Each entry reports the mean and standard deviation of the state discrepancy computed from 4 randomly generated initial guesses and rendering conditions.

## B.3 Training Details

We share the same experiment settings across imitation learning, and visuomotor control experiments experiments. We use RMSProp optimizer with a learning rate of 3e-3 and a momentum of 0.5. We optimize the system parameters or the action parameters by 100 iterations.

## C More Experiments

### C.1 Visuomotor Control

We consider a visuomotor control task defined as follows: given a target image displaying the desired state of the physical system, we optimize a sequence of actions that steer the physical system to the target state from a randomly generated initial state. We set the target states by selecting them from the ground truth in the imitation learning tasks. We report in Table 4 the state error computed from experiments repeated with various rendering configurations and initial guesses. The state error is defined as L1 distances between the desired state and the final state from the simulator. The smaller state error and standard deviation from our methods in Table 4 shows the advantages of our approach over other baselines. As before, the performance from all methods is capped by the oracle, which requires much more knowledge about the ground-truth state.

### C.2 Real-World Experiment

#### C.2.1 Problem Setup

We consider an imitation learning task for the real-world quadrotor. The imitation learning takes as input the intrinsic and extrinsic parameters of the camera, a real-world quadrotor video clip recorded by the camera, empirical values of the quadrotor's system parameters, and the geometry of the quadrotor represented as a mesh file. We aim to recover an action sequence so that the simulated quadrotor resembles the motion in the input video as closely as possible.

#### C.2.2 Experimental Setup

**Video recording** To obtain a real-world video clip, we set up a calibrated camera with known intrinsic and extrinsic parameters. During the whole recording procedure, the camera remains a fixed position and pose. After we obtain the original video clip, we trim out the leading and tailing frames with little motion and generate the pre-processed video clip with total length of 14.4s and frequency of 10Hz.

**Quadrotor** To replicate a potential deployment, we choose a commercial quadrotor that is publicly available to users. We manually control the quadrotor so that it follows a smooth trajectory.

**Motion capture (MoCap) system** To evaluate the performance of various methods, we also build a MoCap system to log the position and rotation of the quadrotor. We attach six reflective markers to the quadrotor so that the MoCap system tracks the local coordinate system spanned by these markers.

Note that we only use MoCap data for test use, it is not necessarily a step in our pipeline, and none of our methods ever used the MoCap data as a dependency.

**Network and dataset details** To be consistent with other experiments, we use the same network architecture for RISP as the **quadrotor** environment. We uniformly sample the position and pose ensuring that the quadrotor is observable on the screen. Every sampled state comes with a different rendering configuration. We generate in total 10k data points and divide them by 8:2 for training and testing.

### C.2.3 IMITATION LEARNING

Here we perform the imitation learning task where we reconstruct the action sequence from the real-world video clip. This task is much more challenging than the simulated ones. First, the real-world physics has unpredictable noises that is impossible to be modeled perfectly by the differentiable simulation. Additionally, the real-world video has unseen appearances including but not limited to the lighting source and the floor texture. It is also noteworthy that the real-world video is exceedingly long which increases the dimension of action sequence dramatically.

In order to solve these challenges, we design a slightly different pipeline of imitation learning for the real-world experiments using RISP. We first run RISP on the input video to generate a target trajectory of the quadrotor. Next, we minimize a loss function defined as the difference between the target trajectory and a simulated trajectory from a differentiable quadrotor simulator, with the action at each video frame as the decision variables. Because this optimization involves many degrees of freedom and a long time horizon, we find a good initial guess crucial. We initialize the action sequence using the output of a handcrafted controller that attempts to follow the target trajectory. Such an initial guess ensures the quadrotor stays in the camera's view and is used by our method and the baselines.

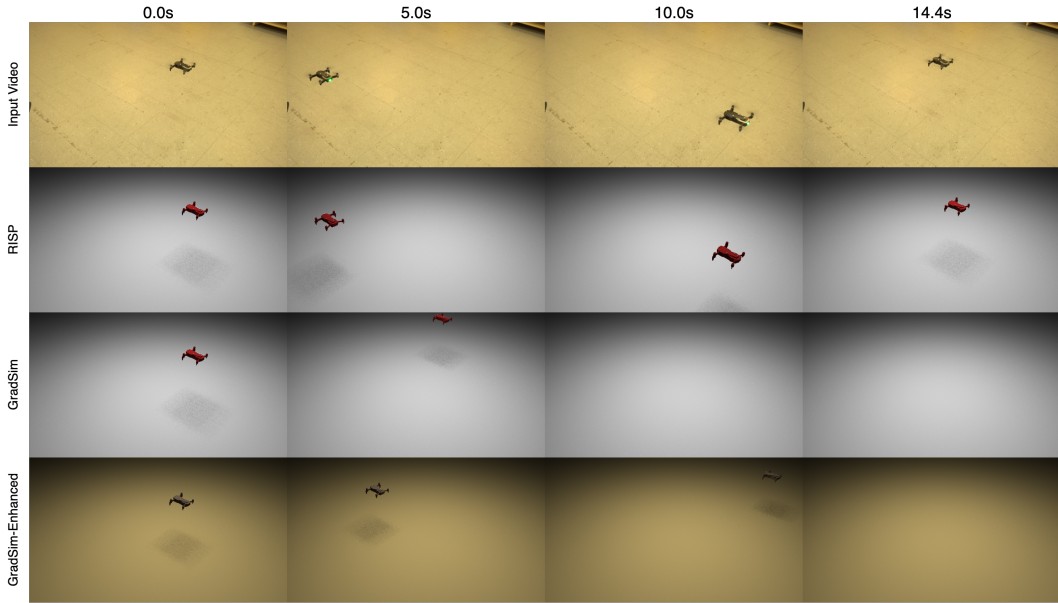

Figure 5: Imitation learning in the real-world experiment. Given a reference video (top), the goal is to reconstruct a sequence of actions that resembles its motion. We illustrate the motions reconstructed using our method (upper middle), the pixelwise loss (lower middle), and its enhanced variant (bottom).

### C.2.4 RESULTS

We show the results of the real-world experiment in Fig. 5 by visualizing a representative subset of frames.

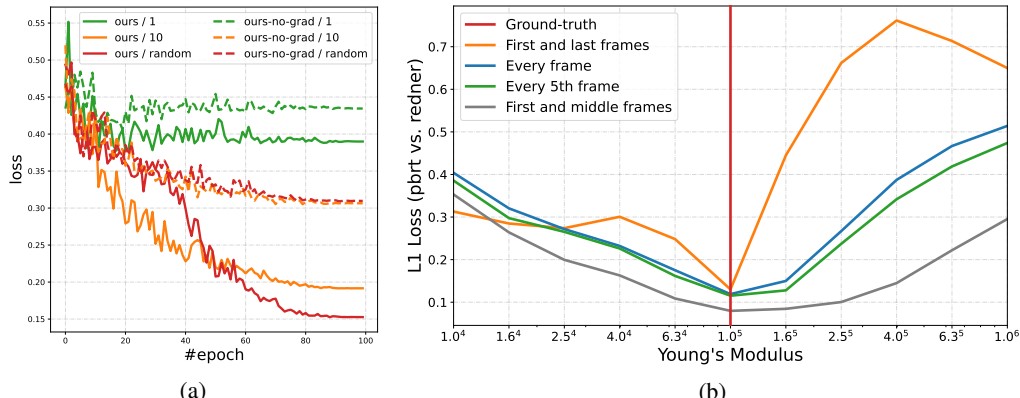

(a)                                                             (b)

Figure 6: **(a)** The number of epoch versus loss curves. The solid and dashed curves represent the training curves of **ours** and **ours-no-grad** respectively. The color of the lines indicates the number of rendering configurations in training set where green and orange are 1 and 10, and red curve samples a different rendering configuration for every training data. **(b)** The loss versus Young's modulus curves. The orange, blue, green, and gray curves represent the loss landscapes with respect to the Young's modulus given different temporal sampling rates. The red vertical line shows where the ground-truth Young's modulus locates.

**RISP**    To validate the robustness of our method, we aggressively randomized the rendering configuration so that it differs significantly from the input video. Our method reconstructs a similar dynamic motion without access to the rendering configuration or the exact dynamic model in the input video.

**GradSim**    We compare the pixelwise loss from $\nabla$Sim (26) with our method by sharing the identical rendering configuration and initial guesses. We observe that the pixelwise loss does not provide valid guidance to the optimization and steers the quadrotor out of the screen immediately. We presume the major reason behind the degeneration of pixelwise loss is the assumption of a known rendering configuration in the target domain. However, such information is generally non-trivial to obtain from a real-world video.

**GradSim-Enhanced**    According to our presumption above, we propose two modifications for improvement: First, we manually tuned the rendering configuration until it appears as close as possible to the real-world video, and second, we provide it with a good initial guess of the action sequence computed based on the ground-truth trajectory recorded from a motion capture system. Note that the good initial guess from motion capture system is generally impossible to access in real-world applications and is not used in **RISP** or **GradSim**. Even with these strong favors, **GradSim-Enhanced** only recovers the very beginning of the action sequence and ends up with an uncontrollable drifting out of screen.

## C.3    ABLATION STUDY

### C.3.1    RENDERING GRADIENTS

We study the impact of rendering gradients on the data efficiency by the comparison between **ours** and **ours-no-grad**. We reuse the state estimation data sets (Table 1) on **quadrotor** but vary the number of rendering configurations between 1 and 10. We then train both of our methods for 100 epochs and report their performances on a test set consisting of 200 randomly states, each of which is augmented by 10 unseen rendering configurations. We show the result on Fig. 6a. The right inset summarizes the performances of **ours** (solid lines) and **ours-no-grad** (dashed lines) under varying number of rendering configurations, with the green, orange, and red colors corresponding to results trained on 1, 10, and randomly sampled rendering configurations. It is obvious to see that all solid lines reach a lower state estimation loss than their dashed counterparts, indicating that our rendering gradient digs more information out of the same amount of rendering configurations. It is worth noting that with only 10 rendering configurations (orange solid line), our method with rendering gradients

achieves a lower loss than the one without but using randomly sampled rendering configurations (red dashed line), which reflects the better data efficiency.

By comparing **ours** and **ours-no-grad** from Table 1, 2, 3, and 4, we can see that having the rendering gradients in our approach is crucial to its substantially better performance. We stress that having a rendering-invariant state estimation is the core source of generalizability in our approach and the key to success in many downstream tasks.

### C.3.2 TEMPORAL SAMPLING RATE

We study the impact of temporal sampling rate on the performance of RISP using four different sampling strategies: sampling densely on every frame, sampling sparsely on every 5th frame, sampling only on the first and last frames, and sampling only on the first and middle frames. We reuse the system identification task in **rod** environment and plot the loss landscapes of them around the ground-truth Young's Modulus in Fig. 6b. We observe that RISP is robust against different sampling rates due to the same global optima of all curves. Additional, except for the extreme case with substantial information loss (first and last frames), most of the curves are unimodal with one local minimum indicating an easier problem in optimization. Thus, we believe it is safe to expect our method with a standard gradient-based optimizer to succeed under various temporal sampling rates.

