# OpenReview forum: "RISP: Rendering-Invariant State Predictor with Differentiable Simulation and Rendering for Cross-Domain Parameter Estimation"
_ICLR.cc/2022/Conference — ICLR 2022 Oral_

### Official Review · Reviewer_qnnY · 2021-10-30

**Correctness:** 4
**Technical Novelty And Significance:** 3
**Empirical Novelty And Significance:** 4
**Recommendation:** 8
**Confidence:** 4

**Main Review:**

This work addressed a crutial problem. The proposed method is novel and neat. The paper is well-written. I like the main idea and technically novel components proposed, such as the rendering invariant regularization term and efficiently calculating the second-order gradient. The results have demonstrated the importance of these components and the improvements over previous methods.
For the weakness of this paper, it is not clear how this method works on real datasets. I suggest that the proposed method be evaluated on more datasets, especially real datasets.

**Summary Of The Paper:**

This paper presents a new method of predicting physics simulation parameters and rendering configurations from an RGB video. Unlike previous methods that calculate the loss function in image space, this work proposes to calculate in the simulation state space to avoid the issues (e.g. being stuck in a local minimum) when the reference video is very different from the generated video. Specifically, a rendering-invariant state-prediction (RISP) network is pretrained with the generated data from a differentiable renderer under various rendering conditions.  The RIST network is then appended to the output of a differentiable renderer to make the pipeline from simulation and rendering parameters to states predicted from images fully differentiable. Besides a state prediction loss term for training this pipeline, a novel regularization term is used to enforce the RISP network rendering invariant. In addition, a new training strategy to efficiently calculate the gradient for the loss function is proposed. The experiments have shown that the proposed method significantly outperforms the state-of-the-arts and the proposed components contribute to the final results with big improvements.

**Summary Of The Review:**

This paper is a solid submission, which solves an important problem and proposes some novel ideas. I expect the authors to evaluate the method on more datasets, especially real datasets.

---

> ### Author Response · Authors · 2021-11-21
> **Response to Reviewer qnnY**
>
> We thank the reviewer for the suggestion.
>
> **Q1:** "I expect the authors to evaluate the method on more datasets, especially real datasets."
>
> **A1:** We agree that applying our approach to real-world problems is definitely an exciting direction. Therefore, we have added a real-world experiment that imitates a quadrotor's motion from a video clip. Please refer to our general response above [[response]](https://openreview.net/forum?id=uSE03demja&noteId=C9ZB2AOsUrw) and our website [[website]](https://sites.google.com/view/risp-iclr-2022/real-quadrotor) for more details.
>
> ---
>
> Reference links:
>
> - general response of the real-world experiment: [https://openreview.net/forum?id=uSE03demja&noteId=C9ZB2AOsUrw](https://openreview.net/forum?id=uSE03demja&noteId=C9ZB2AOsUrw)
> - website of the real-world experiment: [https://sites.google.com/view/risp-iclr-2022/real-quadrotor](https://sites.google.com/view/risp-iclr-2022/real-quadrotor)

---

### Official Review · Reviewer_UwVk · 2021-11-02

**Correctness:** 4
**Technical Novelty And Significance:** 3
**Empirical Novelty And Significance:** 3
**Recommendation:** 8
**Confidence:** 3

**Main Review:**

Strengths:

- The paper is very well written and contributions are clear.
- The idea of using a rendering-invariant state-prediction as a pre-step before the gradsim-like framework is novel and sensible.
- The idea of the using the gradient with respect to the rendering parameters as a regularization is novel and interesting. It could additionally be very used in many other fields where training on synthetic data is used with random rendering parameters.
- The Combination of weak/strong/oracle baselines chosen in the experiments are very appropriate for comparison.
- The paper performs a large range of experiments using several environment and tasks to study the effectiveness of the proposed method compared to baselines.
- The ablation study shows the impact of the gradient loss on the state prediction model performance.

Weaknesses:

- It is not clear if the training and testing rendering environments are the same. This means that the parameter distribution in both training (for the RISP) and testing cases come from the same distribution. In reality, often the test parameter distribution comes from a different distribution. It would be interesting to see results when the train and test rendering parameter distributions are different.
- It would be interesting to see how the various methods would perform on real videos.

**Summary Of The Paper:**

The paper focuses on the problem of estimating dynamic parameters of a physical system from videos under unknown rendering conditions.
It presents a novel idea of using a rendering-invariant state-prediction (RISP) network that predicts the state from a rendered image and can be integrated into a framework with a differentiable simulation and rendering engine for parameter estimation. Training this network is done using domain randomization technique with synthetic data. Additionally, the paper introduces a novel gradient loss that further pushes the network to be invariant to rendering parameters.

**Summary Of The Review:**

The paper presents mainly two new ideas, the RISP network for state estimation and the gradient loss for regularization. Both are novel and can be interesting and impactful for the community and push forward the state-of-the-art in using synthetic data for parameter estimation. The quantitative and qualitative experimental results on synthetic data show clear improvements compared to the previous methods.

---

> ### Author Response · Authors · 2021-11-21
> **Response to Reviewer UwVk**
>
> We thank the reviewer for the valuable feedback and positive comments. We are pleasant that the reviewer found the paper to be novel and interesting.
>
> **Q1:** "It would be interesting to see results when the train and test rendering parameter distributions are different."
>
> **A1:** This is a brilliant question indicating the key problem RISP attempts to solve: adaptation between various rendering configurations. The rendering configurations in our training and test environments are indeed from different distributions. In fact, some of the material models we used in our test environment (pbrt) are not even available in our training environment (mitsuba). Our real-world experiment [[website]](https://sites.google.com/view/risp-iclr-2022/real-quadrotor) is another example of using rendering configurations from different distributions, and RISP still achieves reasonable performance.
>
> ---
>
> **Q2:** "It would be interesting to see how the various methods would perform on real videos."
>
> **A2:** Thank you for this suggestion. We have provided a new real-world experiment. Please find its details in our general response above [[response]](https://openreview.net/forum?id=uSE03demja&noteId=C9ZB2AOsUrw) and in our website [[website]](https://sites.google.com/view/risp-iclr-2022/real-quadrotor).
>
> ---
>
> Reference links:
>
> - general response of the real-world experiment: [https://openreview.net/forum?id=uSE03demja&noteId=C9ZB2AOsUrw](https://openreview.net/forum?id=uSE03demja&noteId=C9ZB2AOsUrw)
> - website of the real-world experiment: [https://sites.google.com/view/risp-iclr-2022/real-quadrotor](https://sites.google.com/view/risp-iclr-2022/real-quadrotor)

---

### Official Review · Reviewer_paRJ · 2021-11-08

**Correctness:** 3
**Technical Novelty And Significance:** 3
**Empirical Novelty And Significance:** 3
**Recommendation:** 8
**Confidence:** 4

**Main Review:**

### Strengths

* **S1** This paper tackles the challenging (inverse) problem of directly recovering object properties (system identification) or control parameters (visuomotor control) from image/video observations. The current best approach to this ($\nabla$Sim) proposes to compose differentiable physics simulation and differentiable rendering to result in a computation graph where ground-truth physical parameters are no longer required. However, this approach has a crucial shortcoming: it works well only when the reference and target trajectories are drawn from the same distribution (i.e., identical physics and rendering engines used across both). This work proposes RISP to bridge that gap (bridging this gap is crucial to enable several downstream, including real-world, applications).

* **S2** The paper is very well-presented. The core ideas are easy to follow, and the rationale for incorporating the rendering gradients into a regularization term is well laid-out. Of the three contributions, I believe this to be the more significant one (as also corroborated by Fig. in section 4.6)

* **S3** This approach (RISP) demonstrates strong performance over current art -- this is over multiple environment settings (rigid, articulated, deformable object identification; control).

In general, this is well-executed work and I would as such recommend acceptance. However, there are a few issues I hope to see discusses/addressed over the rebuttal phase.

### Weaknesses

* **W1** Clarifying the impact of differentiable physics simulation vs rendering: Are the train sets generated using a different differentiable physics engine? (the manuscript mentions they're generated using a different rendering engine -- but, arguably, generating them using a different physics engine as well would create a wider domain gap) This also raises several interesting questions such as: will RISP generalize to scenarios where the underlying dynamics change too? If not, are there other domain randomization / regularization methods that can assist?

* **W2** Baseline choices: Do all baselines use all frames in the reference trajectory to compute the loss. Just as dense pixelwise loss baselines are ablated upon by bringing in other baselines that treat the image as a whole in loss computation; a similar analogy may be drawn at the sequence level (i.e., computing the full pixelwise loss across all frames in a sequence, vs. using pixelwise loss across select frames in a sequence)

* **W3** Disentanglement/Compositionality of the learned RISP: The current training strategies do not seem to explicity focus on disentangling state representations or improving compositionality of the learned representation. (This seems clearly out of scope for the current work, but I'd like to bring this discusison point here to perhaps prompt addition of clarifying statements in the manuscript). Does this have an imact on e.g., where the object may lie within an image, and perhaps impact the extension of RISP to simulations with multiple objects?


### Minor comments

These comments are nitpicks/typos and are easily addressed in a minor revision. The authors needn’t respond to these

* “Articulate-body” -> articulated body
* “Simulates” -> simulate

**Summary Of The Paper:**

This paper proposes a general approach to leveraging differentiable simulator for the downstream tasks of system identification and visuomotor control WITHOUT requiring access to the true underlying states. They do so by predicting a "rendering-invariant state" that results in a much stabler loss landscape and reduces domain gap / mismatch. Experiments over 4 environments indicate the merits of this approach over current state-of-the-art.

**Summary Of The Review:**

This is a well-written paper describing an idea of substantial interest to the physical reasoning and the visual reasoning communitties. I only have clarifying concerns with this work and feel the paper will be well-rounded if the limitations of the work are discussed upfront.

---

> ### Author Response · Authors · 2021-11-11
> **Looking for clarification**
>
> Thank you very much for your positive comments and constructive suggestions. Could you please clarify **W2**? If you question whether all baselines use full trajectory for the sysid and imitation learning tasks, the answer is yes. We believe this is a pretty common setting and also consistent with $\nabla$Sim. Please do not hesitate to contact us if there is any additional clarification or experiment we can offer. Thank you.

---

> > ### Comment · Reviewer_paRJ · 2021-11-15
> > **Clarification on W2**
> >
> > With **W2**, I am trying to get a sense of how dense (in terms of time) the loss needs to be. For instance, if one were to use only a sparse subset of (time-synchronized) frames from within a trajectory (in the limiting case, a single image as used in $\nabla$Sim), it would be interesting to note the impact on performance. For dense pixelwise MSE losses, computing these quantities over an entire trajectory offers worse performance as opposed to computing MSE over just the "first and last" frames, as reported in $\nabla$Sim.

---

> ### Author Response · Authors · 2021-11-21
> **Response to Reviewer paRJ**
>
> We thank the reviewer for providing additional clarifications. Below we provide a detailed response to that as well as other concerns.
>
> **Q1:** "Will RISP generalize to scenarios where the underlying dynamics change too?" (**W1** in the review)
>
> **A1:** This is a great point that our new real-world experiment [[website]](https://sites.google.com/view/risp-iclr-2022/real-quadrotor) can help answer. In that experiment, we not only had no access to the ground-truth system parameters (we used publicly available empirical numbers) but also omitted many real-world quadrotor dynamics in our simulation, e.g., ground effects, motor responses, propeller dynamics, and battery models. Still, our method reconstructed similar motions from the real-world video, showing its potential generalizability among different dynamic models.
>
> ---
>
> **Q2:** "how dense (in terms of time) the loss needs to be." (**W2** in the review)
>
> **A2:** We thank the reviewer for the clarification, and we agree that this ablation study helps better understand how RISP works. We added this ablation study in our appendix using a setup similar to Fig. 12 in the $\nabla$Sim paper. Specifically, we use the system identification task in our **rod** environment and present the loss-vs-Young's modulus curves with various temporal sampling rates. Note that unlike in $\nabla$Sim, the loss is now computed through our RISP network. The figure [[figure]](https://drive.google.com/file/d/1TNe8z3sXVvGlb4QZL_1qoBv__HbtXXSr/view?usp=sharing) shows that changes in temporal sampling rate do not seem to affect the loss landscape substantially: most of these curves, except for the first-and-last-frames one, are unimodal functions with one local minimum. Therefore, we can expect RISP plus a standard gradient-based optimizer to succeed under various temporal sampling rates.
>
> ---
>
> **Q3:** "Disentanglement/Compositionality of the learned RISP" (**W3** in the review)
>
> **A3:** We agree that it would be exciting to introduce disentanglement or compositionality into RISP for better explainability. For example, in the training phase, one might introduce Variational Information Maximization [Barber et al., 2004] into the framework to encourage spontaneous interpretation. Additionally, to analyze a trained RISP, a clustering algorithm might be helpful to understand any emerging behaviors behind the representation. These enhancements and analyses are definitely excellent additions to RISP, which we will leave as future work.
>
> ---
>
> **Q4:** "the paper will be well-rounded if the limitations of the work are discussed upfront."
>
> **A4:** Thank you for the suggestion. We plan to add a limitation section in our manuscript, which we briefly summarize below:
>
> - We require knowledge of the intrinsic and extrinsic parameters of the camera, which is not always accessible in a real-world scenario. We expect this limitation can be resolved easily by incorporating in RISP the gradients for camera parameters, which are available in some modern differentiable renderers.
> - We also require a moderately accurate object geometry. A potential direction is to combine recent advances in Neural Radiance Fields (NeRF) with our approach and infer the geometry without omniscient information of cameras.
> - We require a differentiable simulator that can capture the dynamic model of the object. Such a differentiable simulator may not be available for complex, real-world scenes with intricate dynamics, e.g., fluidic systems, deformable objects with rich contact, or multi-physics systems.
>
> ---
>
> Reference:
>
> - [Barber et al., 2004] D. Barber and F. V. Agakov, "The IM algorithm: A variational approach to information maximization" in NIPS, 2003.
>
> Reference Links:
>
> - website of the real-world experiment: [https://sites.google.com/view/risp-iclr-2022/real-quadrotor](https://sites.google.com/view/risp-iclr-2022/real-quadrotor)
> - figure in A2: [https://drive.google.com/file/d/1TNe8z3sXVvGlb4QZL_1qoBv__HbtXXSr/view?usp=sharing](https://drive.google.com/file/d/1TNe8z3sXVvGlb4QZL_1qoBv__HbtXXSr/view?usp=sharing)

---

### Author Response · Authors · 2021-11-21
**A Real-World Experiment**

**Problem setup**

We consider imitating a quadrotor's motion from a real-world video clip.

- **Input:** A real-world quadrotor video clip, intrinsic and extrinsic parameters of the camera, empirical values of the quadrotor's system parameters (obtained from the vendor's website), and the geometry of the quadrotor represented as a mesh file.
- **Output:** An action sequence so that when applied to the quadrotor in simulation, we expect its motion to resemble the motion in the input video as closely as possible.

---

**Challenges**

Our task requires matching information between simulation and reality, a challenging open problem in computer vision, robotics, and graphics. Therefore, solving this task perfectly would require tremendous efforts:

- The real-world quadrotor contains hard-to-measure parameters and hard-to-model dynamics, e.g., ground effects, motor responses, propeller dynamics, and battery models.
- A real-world video typically contains complex textures, materials, and lighting conditions, requiring substantial manual work to reconstruct them in a realistic renderer.
- The quadrotor's sensing, control, and actuation modules are often polluted by environmental noises, which are challenging to model and infer in a simulated environment.

Despite these challenges, our method achieved a qualitatively good result with a standard differentiable rigid-body simulator and renderer, showing its generalizability across moderate discrepancies in dynamic models and rendering configurations.

---

**Method**

- **Video recording and pre-processing:** We flew a quadrotor in an indoor environment with a clean background and clear lighting. We recorded its motion using a static camera whose parameters were calibrated beforehand. We removed the takeoff and landing motions from the beginning and end of the video and downsampled the video to 360x640 resolution at 10 FPS. The final video is 14.4 seconds long.
- **Training the RISP network:** We trained the network the same way as the virtual **quadrotor** example described in our manuscript before.
- **Solving the task:** We first ran RISP on the input video to generate a target trajectory of the quadrotor. Next, we minimized a loss function defined as the difference between the target trajectory and a simulated trajectory from a differentiable quadrotor simulator, with the action at each video frame as the decision variables. Because this optimization involves many degrees of freedom and a long time horizon, we found a good initial guess crucial. We initialized the action sequence using the output of a handcrafted controller that attempts to follow the target trajectory. Such an initial guess ensures the quadrotor stays in the camera's view and is used by our method and the baselines.

---

**Results**

We show three results on our website [[website]](https://sites.google.com/view/risp-iclr-2022/real-quadrotor): one from our method (**RISP**) and two from the state-of-the-art approach with variations (**GradSim** and **GradSim-Enhanced**). Each result is a side-by-side video comparison between the reconstructed motion and the input video.

- **RISP:** Our method reconstructs a similar dynamic motion without knowing the rendering configuration and the exact dynamic model in the input video.
- **GradSim:** The setup is identical to **RISP** except that the loss function is replaced with the pixel-wise difference defined in the original $\nabla$Sim paper. The reconstructed motion fails to resemble the motion in the input video. We believe one major reason is that **GradSim** assumes the rendering configuration in the target domain is known. However, such information is generally difficult to access from a real-world video.
- **GradSim-Enhanced:** We improved the performance of **GradSim** by 1) manually tuning the renderer's configuration to match the video input as closely as possible and 2) providing a good initial guess of the action sequence computed based on the ground-truth trajectory recorded from a motion capture system. Note that such a good initial guess from motion capture data is not used in **RISP** and **GradSim** results above and is generally inaccessible in real-world applications. With this additional help, **GradSim-Enhanced** manages to mimic the motion at the beginning of the video but still fails to replicate the full trajectory reliably.

We believe that RISP's encouraging result shows the potential of applying differentiable physics plus differentiable rendering techniques in real-world applications. We refer the reviewers to our appendix for a full description of our results and the details of our method in this real-world experiment.

---

Reference link:

- website of the real-world experiment: [https://sites.google.com/view/risp-iclr-2022/real-quadrotor](https://sites.google.com/view/risp-iclr-2022/real-quadrotor)

---

> ### Comment · Reviewer_paRJ · 2021-11-22
> **Real-world experiment adds value**
>
> Dear authors,
>
> Thank you for addressing some of my (minor) concerns with this additional experiment. I believe this additional experiment brings in a lot more value to the paper, particularly as most work dealing with differentiable simulation has required precise dynamics and/or rendering configuration match. This portrays one advantage of RISP in being fairly robust to such configuration changes (while the current result is only qualitative, it is indeed a challenging setting).
>
> One minor remark that could potentially be discussed in a revision: in typical (kinematic) control and planning settings, quadrotors end up being modeled as rigid bodies nonetheless, without considering propeller-blade effects and such. So I believe differentiable simulation can get around by making such an assumption too (which is vindicated in the presented qualitative result).

---

> > ### Author Response · Authors · 2021-11-24
> > **Response to reviewer paRJ**
> >
> > Dear reviewer paRJ,
> >
> > Thank you for your detailed comments. We are glad to see you appreciate our additional real-world experiment. We agree with the suggestion about the discussion to add and will incorporate it into our final manuscript.

---

### Author Response · Authors · 2021-11-21
**General Response**

Dear reviewers,

We thank all reviewers for their feedback and comments on our manuscript. We are glad to see that the reviewers have found that

- the problem we propose to solve is challenging and crucial (reviewer paRJ, qnnY),
- our idea is novel and neat (reviewer UwVk, qnnY),
- the choice of baselines is appropriate (reviewer UwVk),
- our method achieves strong performance (reviewer paRJ, UwVk, qnnY),
- our manuscript is well-written (reviewer paRJ, UwVk, qnnY).

To address concerns raised by the reviewers, we have made the following changes in our work and updated the manuscript (textual revisions marked in red) accordingly:

**Additional experiments**

- We added a real-world experiment [[website]](https://sites.google.com/view/risp-iclr-2022/real-quadrotor) that imitates the trajectory of a quadrotor from a video, which we will elaborate in the next post [[response]](https://openreview.net/forum?id=uSE03demja&noteId=C9ZB2AOsUrw) (reviewer paRJ, UwVk, qnnY).
- We added an ablation study on the influence of temporal sampling rate (reviewer paRJ).

**Clarifications and discussions**

- We discussed the impact of dynamic model discrepancies, the disentanglement/compositionality of RISP, and the limitations of our method (reviewer paRJ).
- We clarified the differences between the training and testing environments (reviewer UwVk).

We hope our responses have addressed all concerns from the reviewers adequately. We thank all reviewers again for their time and feedback, and please feel free to let us know if there are other clarifications or experiments we can offer. We would really appreciate it if the reviewers could consider raising their scores after evaluating our updates.

---

Reference link:

- general response of the real-world experiment: [https://openreview.net/forum?id=uSE03demja&noteId=C9ZB2AOsUrw](https://openreview.net/forum?id=uSE03demja&noteId=C9ZB2AOsUrw)
- website of the real-world experiment: [https://sites.google.com/view/risp-iclr-2022/real-quadrotor](https://sites.google.com/view/risp-iclr-2022/real-quadrotor)

---

### Decision · Program_Chairs · 2022-01-20

**Decision:**

Accept (Oral)

**Comment:**

This paper proposes a method to solve the inverse problem of identifying parameters of a dynamic physical system from image observations. The main idea is to train a rendering-invariant state-prediction (RISP), which estimates the inverse mapping from the pixel to the state domain. The authors introduce a new loss to this end, and an efficient gradient computation of the loss.

The paper received three clear accept recommendations. The reviewers discussed the potential improvement of RISP when combined to disentanglement methods, and also raise several concerns regarding experiments, e.g. rendering conditions during training and testing, or evaluation on real data. The rebuttal did a good job in answering reviewers' concerns, and the reviewers especially appreciated the new results on real videos. Eventually, all reviewers recommended a clear acceptance of the paper.

The AC's own readings confirmed the reviewers' recommendations. The paper is introduces very solid contributions for solving the complex task of physical parameter identification in the unobservable setting. The paper is also clear and well written, and validated with convincing experimental results. Therefore, the AC recommends acceptance.